# ROBUST UNIVERSAL ADVERSARIAL PERTURBATIONS

## ABSTRACT

Universal Adversarial Perturbations (UAPs) are imperceptible, image-agnostic vectors that cause deep neural networks (DNNs) to misclassify inputs from a data distribution with high probability. In practical attack scenarios, adversarial perturbations may undergo transformations such as changes in pixel intensity, rotation, etc. while being added to DNN inputs. Existing methods do not create UAPs robust to these real-world transformations, thereby limiting their applicability in attack scenarios. In this work, we introduce and formulate *robust UAPs*. We build an iterative algorithm using probabilistic robustness bounds and transformations generated by composing arbitrary sub-differentiable transformation functions to construct such robust UAPs. We perform an extensive evaluation on the popular CIFAR-10 and ILSVRC 2012 datasets measuring our UAPs' robustness under a wide range common, real-world transformations such as rotation, contrast changes, etc. Our results show that our method can generate UAPs up to 23% more robust than existing state-of-the-art baselines.

## 1 INTRODUCTION

Deep neural networks (DNNs) have achieved impressive results in many application domains such as natural language processing (Abdel-Hamid et al., 2014; Brown et al., 2020), medicine (Esteva et al., 2017; 2019), and computer vision (Simonyan & Zisserman, 2014; Szegedy et al., 2016). Despite their performance, they can be fragile in the face of adversarial perturbations: small imperceptible changes added to a correctly classified input that make a DNN misclassify. While there is a large amount of work on generating adversarial perturbations (Szegedy et al., 2013; Goodfellow et al., 2014; Moosavi-Dezfooli et al., 2016; Madry et al., 2017; Carlini & Wagner, 2017; Xiao et al., 2018a; Dong et al., 2018; Croce & Hein, 2019; Wang et al., 2019; Zheng et al., 2019; Andriushchenko et al., 2019; Tramèr et al., 2020), the threat model considered by these works cannot be realized in practical scenarios. This is because the threat model depends upon unrealistic assumptions about the power of the attacker: the attacker knows the DNN input in advance, generates input-specific perturbations in real-time and *exactly* combines the perturbation with the input before being processed by the DNN.

**Practically feasible adversarial perturbations.** In this work, we consider a more practical adversary to reveal real-world vulnerabilities of state-of-the-art DNNs. We assume that the attacker (i) does not know the DNN inputs in advance, (ii) can only transmit additive adversarial perturbations, and (iii) their transmitted perturbations are susceptible to modification due to real-world effects. Examples of attacks in our threat model include adding stickers to the cameras for fooling image classifiers (Li et al., 2019b) or transmitting perturbations over the air for deceiving audio classifiers (Li et al., 2019a). Note that this threat model is distinct from directly generating adversarial examples (Athalye et al., 2018) which require access to the original input.

The first two requirements in our threat model can be fulfilled by generating Universal Adversarial Perturbations (UAPs) (Moosavi-Dezfooli et al., 2017). Here the attacker can train a single adversarial perturbation that has a high probability of being adversarial on all inputs in the training distribution. However, as our experimental results show, the generated UAPs need to be combined with the DNN inputs precisely, otherwise they fail to remain adversarial. In practice, changes to UAPs are likely due to real-world effects. For example, the stickers applied to a camera can undergo changes in contrast due to weather conditions or the transmitted perturbation in audio can change due to noise in the transmission channel. This non-robustness reduces the efficiency of practical attacks created with existing methods (Moosavi-Dezfooli et al., 2017; Shafahi et al., 2020; Li et al., 2019b;a).

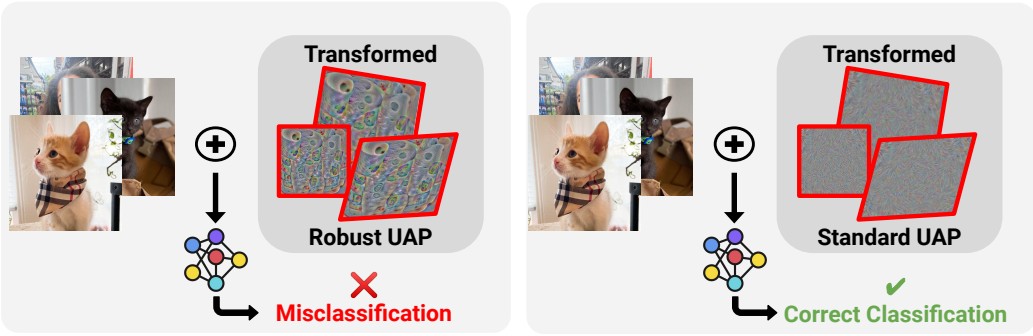

Figure 1: Robust UAPs (left) cause a classier to misclassify on *most* of the data distribution even after transformations are applied on them. Standard UAPs (right) are not robust to transformations and have a low probability of remaining UAPs after transformation.

**This work: Robust UAPs.** To overcome the above limitation, we propose the concept of robust UAPs: perturbations that have a high probability of remaining adversarial on inputs in the training distribution even after applying a set of real-world transformations. The optimization problem in generating robust UAPs (Moosavi-Dezfooli et al., 2017) is the main challenge as we are looking for perturbations that are adversarial for a set of inputs as well as to transformations applied to the perturbations. To address this challenge, we make the following main contributions:

- We introduce *Robust UAPs* and formulate their generation as an optimization problem.

- We design a new method for constructing robust UAPs. Our method is general and works for any transformations generated by composing arbitrary sub-differentiable transformation functions. We provide an algorithm for computing provable probabilistic bounds on the robustness of our UAPs against many practical transformations.

- We perform an extensive evaluation of the effectiveness of our method, `RobustUAP`, on state-of-the-art models for the popular CIFAR-10 (Krizhevsky et al., 2009) and ILSVRC 2012 (Deng et al., 2009) datasets. We compare the robustness of our UAPs under compositions of challenging real-world transformations, such as rotation, contrast change, etc. We show that on both datasets, the UAPs generated by `RobustUAP` are significantly more robust, achieving up to 23% more robustness, than the UAPs generated from the baselines.

Our work is complementary to the development of real-world attacks (Li et al., 2019a;b) in various domains, which require modeling how the universal perturbations change during transmission. `RobustUAP` can improve the efficiency of such attacks by constructing perturbations that are more robust against domain-specific, real-world transformations than possible with existing algorithms (Moosavi-Dezfooli et al., 2017; Shafahi et al., 2020; Li et al., 2019a;b).

## 2 BACKGROUND

In this section, we provide necessary background definitions and notation for our work.

**Adversarial Examples and Perturbations.** An adversarial example is a misclassified data point that is *close* (in some norm) to a correctly classified data point (Goodfellow et al., 2014; Madry et al., 2017; Carlini & Wagner, 2017). Let $\mu \subset \mathbb{R}^d$ be the input data distribution, $\mathbf{x} \in \mu$ be an input point with the corresponding true label $y \in \mathbb{R}$, and $f : \mathbb{R}^d \to \mathbb{R}^{d'}$ be our target classifier. For ease of notation, we define $f_k(\mathbf{x})$ to be the $k^{\text{th}}$ element of $f(\mathbf{x})$ and allow $\hat{f}(\mathbf{x}) = \arg\max_k f_k(\mathbf{x})$ to directly refer to the classification label. We use $\mathbf{v}$ to reference image specific perturbations and $\mathbf{u}$ to reference universal adversarial perturbations, $\mathbf{v_r}$ and $\mathbf{u_r}$ refer to the robust variants and will be defined in Sec. 3. We now formally define an adversarial example.

**Definition 2.1.** Given a correctly classified point $\mathbf{x}$, a distance function $d(\cdot, \cdot) : \mathbb{R}^d \times \mathbb{R}^d \to \mathbb{R}$, and bound $\epsilon \in \mathbb{R}$, $\mathbf{x}'$ is an *adversarial example* iff $d(\mathbf{x}', \mathbf{x}) < \epsilon$ and $\hat{f}(\mathbf{x}') \neq y$.

In this paper, we consider examples $\mathbf{x}'$ generated as $\mathbf{x}' = \mathbf{x} + \mathbf{v}$ where $\mathbf{v}$ is an *adversarial perturbation*.

**Universal Adversarial Perturbations.** UAPs are single vector, input-agnostic perturbations (Moosavi-Dezfooli et al., 2017). They differ from traditional adversarial attacks, which create perturbations dependent on each input sample. To measure UAP performance, we introduce the notion of universal adversarial success rate, which measures the probability that a perturbation $\mathbf{u}$ when added to $\mathbf{x}$, sampled from $\mu$, causes a change in classification under $f$.

**Definition 2.2.** Given a data distribution $\mu$, and perturbation $\mathbf{u}$, *universal adversarial success rate* $\text{ASR}_U$ for $\mathbf{u}$, is defined as

$$\text{ASR}_U(f, \mu, \mathbf{u}) = \underset{\mathbf{x} \sim \mu}{P} (\hat{f}(\mathbf{x} + \mathbf{u}) \neq \hat{f}(\mathbf{x})) \tag{1}$$

Using Definition 2.2, we formally define a UAP.

**Definition 2.3.** A *universal adversarial perturbation* is a vector $\mathbf{u} \in \mathbb{R}^d$ which, when added to almost all datapoints in $\mu$ causes the classifier $f$ to misclassify. Formally, given $\gamma$, a bound on universal ASR, and $l_p$-norm with corresponding bound $\epsilon$, $\mathbf{u}$ is a UAP iff $\text{ASR}_U(f, \mu, \mathbf{u}) > \gamma$ and $||\mathbf{u}||_p < \epsilon$.

In general, if the additive perturbations have small $l_p$-norm , then they look like noise and do not affect the semantic content of the image. For ease of notation in later parts of the paper, we can also pose the construction of UAPs as an expectation minimization problem:

$$\underset{u}{\arg\min} \, \mathbb{E}_{\mathbf{x} \sim \mu}[\delta(\hat{f}(\mathbf{x} + \mathbf{u}), \hat{f}(\mathbf{x}))] \text{ s.t. } ||\mathbf{u}||_p < \epsilon \tag{2}$$

where $\delta$ is the Kronecker Delta function (Agarwal, 2013).

## 3 ROBUST UNIVERSAL ADVERSARIAL PERTURBATIONS

In this section, we will define the optimization problem for generating robust UAPs. We first define transformation sets and neighborhoods.

**Definition 3.1.** A *transformation*, $\tau$, is a composition of bijective sub-differentiable transformation functions. A *transformation set*, $T$, is a set of distinct transformations. A point $\mathbf{v}'$ is in the *neighborhood* $N_T(\mathbf{v})$, of $\mathbf{v}$, if there is a transform in $T$ that maps $\mathbf{v}$ to $\mathbf{v}'$. Formally,

$$\mathbf{v}' \in N_T(\mathbf{v}) \iff \exists \tau \in T \text{ s.t. } \tau(\mathbf{v}) = \mathbf{v}' \tag{3}$$

**Example 3.2.** Let $T$ be the set of all transformations represented by a rotation of $\pm 30°$, scaling of up to a factor of 2, and a translation of up to $\pm 2$ pixels, in this case one $\tau \in T$ could be {rotation of $8°$, scaling a factor of 1.2, and translation of -1.3} in that order and $N_T(\mathbf{v})$ would include any point that can be obtained by applying one of the transformations from $T$ on $\mathbf{v}$.

In order to define robust UAPs we introduce robust universal adversarial success rate.

**Definition 3.3.** Given a data distribution $\mu$, transformation set $T$, universal ASR level $\gamma$, bound $\epsilon$ on $l_p$-norm, and perturbation $\mathbf{u_r}$, *robust universal adversarial success rate*, $\text{ASR}_R$, is defined as,

$$\text{ASR}_R(f, \mu, T, \gamma, \mathbf{u_r}) = \underset{\mathbf{u_r'} \sim N_T(\mathbf{u_r})}{P} (\text{ASR}_U(f, \mu, \mathbf{u_r'}) > \gamma \wedge ||\mathbf{u_r'}||_p < \epsilon) \tag{4}$$

The *robust universal adversarial success rate* measures the probability that a neighbor of $\mathbf{u_r}$ is also an UAP on $\mu$, i.e. after transformation it maintains high universal ASR. We note that even though $||\mathbf{u_r}||_p \leq \epsilon$, it can happen that a $\mathbf{u_r'} \in N_T(\mathbf{u_r})$ has $||\mathbf{u_r'}||_p > \epsilon$, this is particularly true for the semantic transformations considered in this work. Therefore, we require that the norm of $\mathbf{u_r'}$ is small.

Using Definition 3.3 we can now formally define a robust UAP.

**Definition 3.4.** A *robust universal adversarial perturbation*, $\mathbf{u_r}$, is one which *most* points within a neighborhood of $\mathbf{u_r}$ when added to *most* points in $\mu$ fool the classifier, $f$. $\mathbf{u_r}$ satisfies $||\mathbf{u_r}||_p < \epsilon$ and $\text{ASR}_R(f, \mu, T, \gamma, \mathbf{u_r}) > \zeta$.

In order to construct robust UAPs, we can pose the following expectation minimization problem:

$$\underset{\mathbf{u_r}}{\arg\min} \, \underset{\mathbf{u_r'} \in N_T(\mathbf{u_r})}{\mathbb{E}} [I(||\mathbf{u_r'}|| < \epsilon) \times \underset{\mathbf{x} \sim \mu}{\mathbb{E}} [\delta(\hat{f}(\mathbf{x} + \mathbf{u_r'}), \hat{f}(\mathbf{x}))]] \text{ s.t. } ||\mathbf{u_r}||_p < \epsilon \tag{5}$$

Here $I : \mathbb{R}^d \to \mathbb{R}$ denotes an indicator function. The inner expectation represents the UAP condition for the transformed perturbation $\mathbf{u_r'}$ while the outer expectation represents the neighborhood robustness condition. Solving Equation 5 requires computing $\mathbf{u_r}$ which minimizes the expectation over the transformation set and data distribution. This composition makes it computationally harder than minimizing over only the transformation set, as in EOT (Athalye et al., 2018), or than minimizing over only the data distribution, as done for standard UAP (Moosavi-Dezfooli et al., 2017).

## 4 GENERATING ROBUST UNIVERSAL ADVERSARIAL PERTURBATIONS

In this section, we will discuss our approach for optimizing Equation 5 to generate UAPs robust to transformations generated by a composition of arbitrary sub-differentiable transformation functions. At a high level, the objective can be seen as gluing the outer expectation, a EOT objective over the transformations applied on the perturbation, with the inner expectation, a UAP objective over the input data distribution. We first describe intuitive baselines for optimizing Equation 5 and then present our new algorithm, `RobustUAP`.

### 4.1 STOCHASTIC GRADIENT DESCENT

The first baseline directly solves Equation 5 using gradient descent. Since we are solving a constrained optimization problem, we cannot use gradient descent directly. Instead, we can solve the Lagrangian-relaxed form of the problem as in (Carlini & Wagner, 2017; Athalye et al., 2018).

$$\underset{\mathbf{u_r}}{\arg\min} \; \underset{\mathbf{u_r'} \in N_T(\mathbf{u_r})}{\mathbb{E}} [I(||\mathbf{u_r'}|| < \epsilon) \times \underset{\mathbf{x} \sim \mu}{\mathbb{E}} [\delta(\hat{f}(\mathbf{x} + \mathbf{u_r'}), \hat{f}(\mathbf{x}))]] - \lambda ||\mathbf{u_r}||_p \quad (6)$$

We use a momentum based Stochastic Gradient Descent (SGD) method for solving Equation 6. Shafahi et al. (2020) suggests that this is an effective method for generating standard UAPs. In order to implement this, we replace the Kronecker Delta function with a loss function, $L$. We iteratively converge towards the inner expectation by computing it in batches, and towards the outer expectation by sampling a large number of transformations. Given that we would like to estimate on a batch, $\hat{\mathbf{x}} \subset \mu$, and a random set of transformations sampled from $T$, $\hat{\tau} \subset T$, we can approximate Equation 6:

$$\frac{I(||\hat{\tau}_j(\mathbf{u_r})|| < \epsilon)}{|\hat{\mathbf{x}}| \times |\hat{\tau}|} \sum_{i=1}^{|\hat{\mathbf{x}}|} \sum_{j=1}^{|\hat{\tau}|} L[f(\hat{\mathbf{x_i}} + \hat{\tau}_j(\mathbf{u_r})), f(\hat{\mathbf{x_i}})] - \lambda ||\mathbf{u_r}||_p \quad (7)$$

Our final algorithm is in Appendix C.

### 4.2 STANDARD UAP ALGORITHM WITH ROBUST ADVERSARIAL PERTURBATIONS

For our second baseline, we leverage the standard UAP algorithm from Moosavi-Dezfooli et al. (2017) (see Appendix D for the algorithm). The standard UAP algorithm iterates over the entire training dataset and at each input, $\mathbf{x_i}$, computes the smallest additive change, $\Delta\mathbf{u}$, to the current perturbation, $\mathbf{u}$, that would make $\mathbf{u} + \Delta\mathbf{u}$ an adversarial perturbation for $\mathbf{x_i}$. Intuitively, over time the algorithm will approach a perturbation that works on most inputs in the training dataset. This approach works by computing robust adversarial perturbations rather than standard adversarial perturbations. At each point $\mathbf{x_i}$, we compute the smallest additive change, $\Delta\mathbf{u_r}$, to the current robust adversarial perturbation, $\mathbf{u_r}$, that would make $\mathbf{u_r} + \Delta\mathbf{u_r}$ a robust adversarial perturbation for $\mathbf{x_i}$.

We search for robust adversarial perturbations by optimizing the expectation that a point in the neighborhood of $\mathbf{v_r}$ is adversarial while restricting the perturbation to an $l_p$ norm of $\epsilon$. We formulate this as the following minimization problem:

$$\underset{\mathbf{v_r}}{\arg\min} \; \underset{\mathbf{v_r'} \in N_T(\mathbf{v_r})}{\mathbb{E}} [I(||\mathbf{v_r'}|| < \epsilon) \times \delta(\hat{f}(\mathbf{x} + \mathbf{v_r'}), \hat{f}(\mathbf{x}))] \text{ s.t. } ||\mathbf{v_r}||_p < \epsilon \quad (8)$$

### 4.3 ROBUST UAP ALGORITHM

The baseline algorithms have two fundamental limitations: (i) they rely on random sampling over the symbolic transformation region, but the sampling strategy does not explicitly try to maximize the robustness of the generated UAP over the entire symbolic region, and (ii) they do not estimate

robustness on unsampled transformations. As a result, the baselines yield suboptimal UAPs (as confirmed by our experiments below). To overcome these fundamental limitations, we create a method to compute probabilistic bounds for expected robustness on an entire symbolic region. We leverage this method for approximating expected robustness in a new algorithm to generate robust UAPs with guarantees. We make a simplifying assumption that $N_T(\mathbf{u_r})$ has a well defined, sampleable probability density function (PDF) as we cannot bound robustness for arbitrary transformations. Our experiments show that even though our assumptions do not hold for all the transformation sets considered in this work, they significantly improve the robustness of our generated UAPs. Our approximation of the expected robustness relies on the following theoretical result:

**Theorem 4.1.** *Given a perturbation* $\mathbf{u_r}$, *a neural network* $f$, *a finite set of inputs* $\mathbf{X}$, *a set of transformations* $T$, *and minimum universal adversarial success rate* $\gamma \in \mathbb{R}$. *Let* $p(\gamma) = P_{\mathbf{u'_r} \sim N_T(\mathbf{u_r})}(ASR_U(f, \mathbf{X}, \mathbf{u'_r}) > \gamma)$. *For* $i \in 1 \ldots n$, *let* $\mathbf{u_r^i} \sim N_T(\mathbf{u_r})$ *be random variables with a well defined PDF and* $I : \mathbb{R}^d \to \mathbb{R}$ *be the indicator function, let*

$$\hat{p}_n(\gamma) = \frac{1}{n} \sum_{i=1}^{n} I(ASR_U(f, \mathbf{X}, \mathbf{u_r^i}) > \gamma) \tag{9}$$

*For accuracy level,* $\psi \in (0, 1)$, *and confidence,* $\phi \in (0, 1)$, *where* $(0, 1)$ *is the open interval between* $0$ *and* $1$. *If* $n \geq \frac{1}{2\psi^2} \ln \frac{2}{\phi}$ *then*

$$P(|\hat{p}_n(\gamma) - p(\gamma)| < \psi) \geq 1 - \phi \tag{10}$$

*Proof.* The bound on $n$ is derived via the Chernoff inequality applied to $\hat{p}_n(\gamma)$ and $\mathbb{E}[\hat{p}_n(\gamma)] = p(\gamma)$ (Chernoff, 1952; Alippi, 2014). Equation 10 holds since computing universal ASR is Lebesgue measurable over the data distribution and since we assume $N_T(\mathbf{u_r})$ has a well defined PDF. □

Theorem 4.1 states that with enough samples from the neighborhood of a perturbation, $\mathbf{u_r}$, the adversarial success rate of $\mathbf{u_r}$ on the entire neighborhood is arbitrarily close to the adversarial success rate of $\mathbf{u_r}$ on sampled transformations with probability greater than $1 - \phi$. One key observation is that the Chernoff bound is independent of the dimensionality of the sample space which allows us to efficiently apply this result to high-dimensional transformation set provided they have a well-defined PDF (e.g., $L_\infty$-ball) and obtain provable bounds on the expected robustness. For the combinations of semantic transformations, such as rotation, translation, etc. used in the experiment section the neighborhood does not have a well-defined PDF, thus we uniformly sample the parameter space of each transformation to produce a point in the neighborhood. We believe uniformly sampling the parameter space is a realistic approximation of real-world effects.

Leveraging Theorem 4.1, we create `EstimateRobustness` which given accuracy, $\psi$, and confidence, $\phi$, returns the robust adversarial success rate on a finite set of inputs with probabilistic robustness guarantees under the assumptions of Theorem 4.1. The pseudocode for `EstimateRobustness` is in Algorithm 1

---
**Algorithm 1** EstimateRobustness

---
1: Draw $n = \lceil \frac{1}{2\psi^2} \ln \frac{2}{\phi} \rceil$ samples $\tau_i \sim T$
2: Compute $\hat{p}_n(\gamma) = \frac{1}{n} \sum_i^n I(ASR_U(f, \mathbf{X}, \tau_i(\mathbf{u_r})) > \gamma)$
3: **Return** $\hat{p}_n(\gamma)$

---

**Our algorithm: `RobustUAP`.** We leverage Theorem 4.1 and Algorithm 1 to develop `RobustUAP`, the pseudocode for which is seen in Algorithm 2. Similar to the SGD baseline, we approximate the expectation in Equation 5 in batches. We start by sampling transformations from the PDF of the neighborhood. We set the number of transformations, $n$, based on Theorem 4.1 to satisfy the desired confidence level and accuracy. For each gradient step, we compute the mean loss over the current batch and set of sampled transforms (line 8). For each set of batch and sampled transformations, instead of making a single gradient update like SGD, we use Projected Gradient Descent (PGD) to iteratively compute a more robust update to the universal perturbation and end only when the estimated robustness on the batch satisfies a given threshold (line 10). At the end of each epoch, we check the robustness across the entire training set and transformation space using `EstimateRobustness` and stop when we have reached the desired performance (line 14).

---

**Algorithm 2** Robust UAP Algorithm

---

1: Initialize $\mathbf{u_r} \leftarrow 0, n \leftarrow \lceil \frac{1}{2\psi^2} \ln \frac{2}{\phi} \rceil$
2: For $i = 1 \dots n$ sample $\tau_i \sim T$
3: **repeat**
4:    **for** $\mathbf{B} \subset \mathbf{X}$ **do**
5:      **if** `EstimateRobustness`$(f, \mathbf{B}, T, \gamma, \mathbf{u_r}, \psi, \phi) < \zeta$ **then**
6:        $\Delta\mathbf{u_r} \leftarrow 0$
7:        **repeat**
8:          Compute $L_{\mathbf{B},\tau} = \frac{1}{|\mathbf{B}| \times n} \sum_{i=1}^{|\mathbf{B}|} \sum_{j=1}^{n} L[f(\mathbf{B_i} + \tau_j(\mathbf{u_r} + \Delta\mathbf{u_r})), f(\mathbf{B_i})]$
9:          $\Delta\mathbf{u_r} = \mathcal{P}_{p,\epsilon}(\Delta\mathbf{u_r} + \alpha \text{sign}(\nabla L_{\mathbf{B},\tau}))$
10:        **until** `EstimateRobustness`$(f, \mathbf{B}, T, \gamma, \mathbf{u_r} + \Delta\mathbf{u_r}, \psi, \phi) < \zeta$
11:        Update the perturbation with projection: $\mathbf{u_r} \leftarrow \mathcal{P}_{p,\epsilon}(\mathbf{u_r} + \Delta\mathbf{u_r})$
12:      **end if**
13:    **end for**
14: **until** `EstimateRobustness`$(f, \mathbf{X}, T, \gamma, \mathbf{u_r}, \psi, \phi) < \zeta$

---

## 5 EVALUATION

Our `RobustUAP` framework is applicable to all transformation sets in a variety of domains. We empirically evaluate our method `RobustUAP` and three baseline approaches (`SGD`, `StandardUAP_RP`, `StandardUAP` (Moosavi-Dezfooli et al., 2017)) on popular models from the vision domain. We show that `RobustUAP` is more robust on both uniform random noise and compositions of real-world transformations such as rotation, scaling, etc.

**Experimental evaluation.** We consider two popular image recognition datasets: CIFAR-10(Krizhevsky et al., 2009) and ILSVRC 2012(Deng et al., 2009). For CIFAR-10, we evaluate on the entire test set (1,000 images) and use a state-of-the-art pretrained VGG16 (Simonyan & Zisserman, 2014) network as the target classification model. For ILSVRC 2012, we evaluate on a random subset of the test set (1,000 images), and use a state-of-the-art Inception-v3 (Szegedy et al., 2016) network. We evaluate the robustness against uniform random noise as well as a composition of transformations from brightness/contrast, rotation, scaling, shearing, and translation. All experiments were performed on a desktop PC with a GeForce RTX(TM) 3090 GPU and a 16-core Intel(R) Core(TM) i9-9900KS CPU @ 4.00GHz.

We report the results for $l_2$-norm with $\epsilon = 100$ for ILSVRC 2012 and $\epsilon = 10$ for CIFAR-10. These values were chosen based on the values presented by the original UAP paper (Moosavi-Dezfooli et al., 2017). We use an image normalization function given by our pretrained models and thus scaled our $\epsilon$ values accordingly. We note that the $\epsilon$-values are significantly smaller than the image norms. Therefore the generated perturbation is imperceptible and does not affect the semantic content of the image. Due to the hardness of the optimization problem, for the same norm value, the effectiveness of a UAP is less than input-specific perturbations. We note that crafting input-specific perturbations requires making unrealistic assumptions about the power of the attacker as mentioned in the introduction and therefore we do not consider them part of our threat model which aims to generate practically feasible perturbations. We use $\psi = 0.05$ and $\phi = 0.05$ resulting in $n = 738$ for generating samples for our `RobustUAP` algorithm as well as reporting robust ASR in our evaluation. The UAPs are trained on 2,000 images, other parameters for evaluation are given in Appendix E.

### 5.1 ROBUSTNESS TO RANDOM NOISE

First, we show that our algorithm generates UAPs robust against uniform random noise. Here our neighborhood is defined as an $L_\infty$ ball of radius $\epsilon$ around the perturbation. $U(\epsilon)$ represents noise drawn uniformly from such a ball. Figure 2 shows the performance of each algorithm. For example, the `RobustUAP` algorithm achieves a $\text{ASR}_U$ of 0.9 greater than 97% of the time under $U(0.1)$ on CIFAR-10, where all other algorithms achieve 0.9 at most 30% of the time. `RobustUAP` outperforms all other algorithms for both noise sizes. `StandardUAP` has a lower mean and higher variance in universal ASR and is much less robust to transformation. A table of Robust ASR results for $\gamma = 0.8$ can be seen in Appendix F. Our Robust ASR results are guaranteed to be $\pm 0.05$ from the

actual result with a probability of $95\%$. For example, we estimate that `RobustUAP` has $\text{ASR}_R$ of $96.1\%$ for U(0.3), we are guaranteed that the true robustness is $> 91.1\%$ with a probability of $95\%$. Note that we get robustness guarantees from `EstimateRobustness` as our neighborhood has a well-defined PDF.

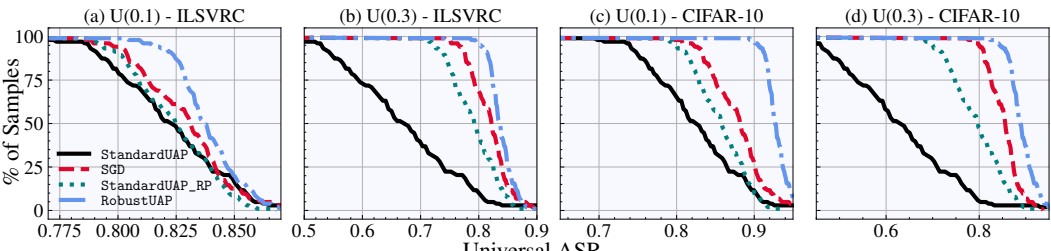

Figure 2: For each method, a point $(x, y)$ in the corresponding line represents the percentage of sampled UAPs ($y\%$) with Universal ASR $> x$ for $U(0.1)$ and $U(0.3)$ on ILSVRC and CIFAR-10.

## 5.2 ROBUSTNESS TO SEMANTIC TRANSFORMATIONS

Next, we consider transformation sets generated by composing five popular semantic transformations in existing literature (Athalye et al., 2018; Balunović et al., 2019): brightness/contrast, rotation, scaling, shearing, and translation.

We use a variety of different compositions to show that our algorithm works under different conditions, and base our parameters for the transformations on (Balunović et al., 2019). For our experiments, $R(\theta)$ corresponds to rotations with angles between $\pm\theta$; $T(x, y)$, to translations of $\pm x$ horizontally and $\pm y$ vertically; $Sc(p)$ to scaling the image between $\pm p\%$; $Sh(m)$ to shearing by shearing factor between $\pm m\%$; and $B(\alpha, \beta)$ to changes in contrast between $\pm\alpha\%$ and brightness between $\pm\beta$. Further details about these transformations can be seen in Appendix A. We consider compositions of different subsets and ranges of these transformations shown in Table 1 including composing all transformations together. The hardness of generating robust UAPs depends on the effect that the transformation set has on the UAP (i.e. random noise has a relatively small effect compared to rotation). The hardness also increases with the number of transformations in the composition as well as the range of parameters for each individual transformation. For example, generating robust UAPs is harder for the composition shown in the first and last row for ILSVRC 2012 in Table 1 compared to the second and third row. The same is true for generating a UAP robust to uniform random noise.

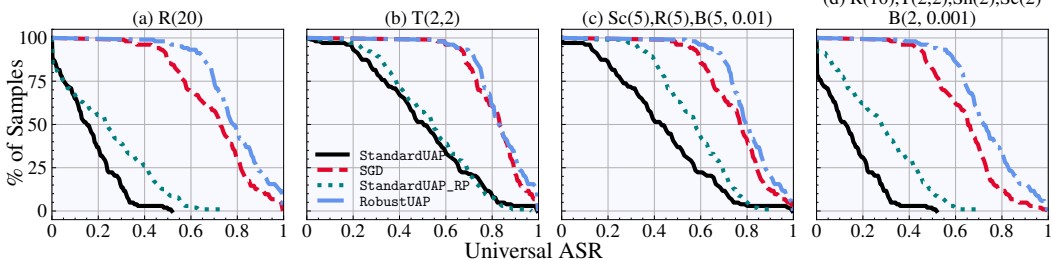

Figure 3: For each method, a point $(x, y)$ in the corresponding line represents the percentage of sampled UAPs ($y\%$) with Universal ASR $> x$ for the different semantic transformations on ILSVRC.

**Robust ASR ($\text{ASR}_R$).** Figure 3 shows performance of UAPs obtained by applying 738 randomly sampled transformations to the original UAPs generated by different methods on ILSVRC, similar graphs for CIFAR-10 can be found in Appendix G. The `RobustUAP` algorithm outperforms all others in each case, we observe that for these harder transformation sets `StandardUAP` loses its effectiveness completely. In Table 1 we compare robust universal adversarial success rate $\text{ASR}_R$ with $\gamma = 0.6$, in other words, we are finding the percentage of sampled neighbors of the perturbation that are still UAPs with $60\%$ effectiveness on the testing set. We provide average $\text{ASR}_U$ scores as well as $\text{ASR}_R$ for different $\gamma$ levels in Appendix H.

Our `RobustUAP` algorithm achieves at least $53.4\%$ higher robust ASR when compared to the standard UAP algorithm on both datasets and the challenging transformation sets shown in Table 1. Furthermore, our `RobustUAP` algorithm significantly outperforms both robust baseline approaches. Except for the $T(2, 2)$ case which we observe to be the easiest, `RobustUAP` achieves at least $11.6\%$ performance gain over the baselines. `SGD` is the best performing baseline and achieves high robust ASR on relatively easier transformation sets performing within $1\%$ of `RobustUAP` on $T(2, 2)$. On harder transformation sets, this gap widens considerably, see Table 1.

| Dataset | Transformation Set | Standard UAP | SGD | Standard UAP_RP | Robust UAP |
|---|---|---|---|---|---|
| ILSVRC 2012 | $R(20)$ | $0.0\%$ | $69.9\%$ | $2.9\%$ | $\mathbf{93.2}\%$ |
| | $T(2, 2)$ | $35.9\%$ | $96.1\%$ | $38.8\%$ | $\mathbf{97.1}\%$ |
| | $Sc(5), R(5), B(5, 0.01)$ | $22.3\%$ | $85.4\%$ | $43.7\%$ | $\mathbf{96.1}\%$ |
| | $R(10), T(2, 2), Sh(2), Sc(2), B(2, 0.001)$ | $0.0\%$ | $63.1\%$ | $2.9\%$ | $\mathbf{86.4}\%$ |
| CIFAR-10 | $R(30), B(2, 0.001)$ | $0.0\%$ | $64.1\%$ | $2.9\%$ | $\mathbf{75.7}\%$ |
| | $R(2), Sh(2)$ | $42.7\%$ | $88.3\%$ | $52.4\%$ | $\mathbf{96.1}\%$ |
| | $R(10), T(2, 2), Sh(2), Sc(2), B(2, 0.001)$ | $0.0\%$ | $58.3\%$ | $7.8\%$ | $\mathbf{79.6}\%$ |

Table 1: Robust ASR of `RobustUAP` compared to the three baselines.

**Visualization.** We visualize UAPs generated with `RobustUAP` and `StandardUAP` transformed with random transformations from $R(10), T(2, 2), Sh(2), Sc(2), B(2, 0.001)$ and added to images in ILSVRC 2012 in Figure 4. Our robust UAPs have a similar level of imperceptibility to standard UAPs and do not affect the semantic content of the images. Robust UAPs affect the model classification after transformation with high probability, unlike standard UAPs.

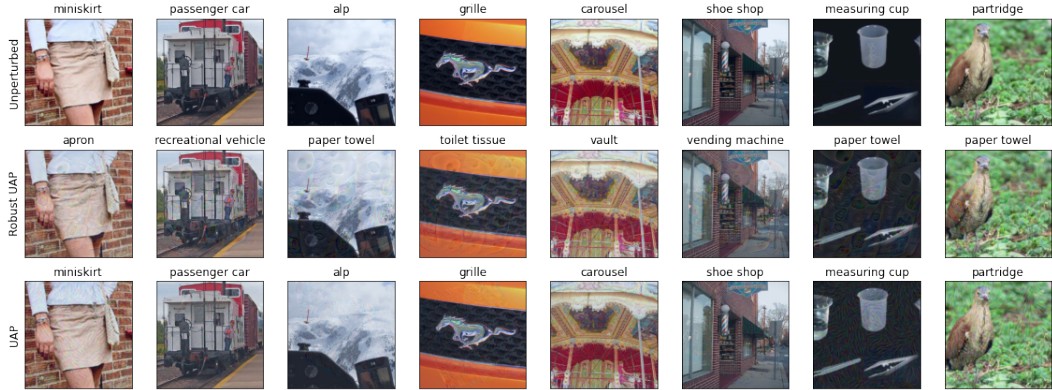

Figure 4: Examples of perturbed images with labels. The top row is unperturbed ILSVRC 2012 test set images, the second row has a randomly transformed robust UAP added to it, and the bottom row has a randomly transformed standard UAP added to it. Labels calculated using Inception-v3.

We further visualize UAPs generated with our three robust algorithms on the same transformation set against a standard UAP generated on ILSVRC 2012 in Figure 5. We observe that UAPs generated by the `StandardUAP` algorithm resemble those generated by the `StandardUAP_RP` algorithm. We believe that this is due to the similarity in the workings of both algorithms. However, the two UAPs are not identical. Under our transformation set the center of the image is least likely to be perturbed so we observe `StandardUAP_RP` algorithm concentrates its budget towards the center. Both the `RobustUAP` and the `SGD` algorithm generate larger patterns distributed over the entire image.

## 5.3 Additional Experiments

In Appendix I we show how our robust UAPs compare to standard UAPs on the non-robust universal ASR metric. In Appendix J, we evaluate our methods on ResNet18 (He et al., 2015) and MobileNet (Howard et al., 2017) for CIFAR-10 and ILSVRC 2012 respectively. The results follow the same trends as those reported in Table 1. In Appendix H we provide the average $\text{ASR}_U$ achieved by all the algorithms and also provide $\text{ASR}_R$ computed with different values of $\gamma$ for the same transformation sets in Table 1. Finally, we provide runtimes for all algorithms in Appendix L.



Figure 5: Comparison of UAPs generated with (a) `StandardUAP`, (b) `RobustUAP`, (c) `StandardUAP_RP`, and (d) `RobustUAP` on ILSVRC 2012.

## 6 RELATED WORK

In this section, we survey works closely related ours.

**UAP Algorithms.** Most works focusing on UAPs (Moosavi-Dezfooli et al., 2017; Mopuri et al., 2018; Zhang et al., 2020a; Khrulkov & Oseledets, 2018; Li et al., 2020; Akhtar et al., 2018; Hendrik Metzen et al., 2017; Zhang et al., 2020b) generate singular vectors and do not consider perturbation robustness. Bahramali et al. (2021) introduces a perturbation generator model (PGM) for the wireless domain which creates UAPs with random trigger patterns. They show that both adversarial training and noise subtracting defenses used in the wireless domain are highly effective in mitigating the effects of a single vector UAP attack; they further show that their method of generating a set of UAPs is an effective way for an attacker to circumvent these defenses. Although PGM provides a method for efficiently sampling unique UAPs, they do not train to be robust to real-world transformations. In contrast, our method enables efficient sampling of UAPs that are robust to transformations.

**Robust Adversarial Examples.** The following papers introduce notions of robustness under different viewpoints and environmental conditions for constructing realizable adversarial examples. This is a different threat model compared to the additive perturbations discussed in this paper. Luo et al. (2018) constructs adversarial examples which minimize human detectability, further introducing the idea of robustness for adversarial examples. They show that their attacks are robust against jpeg compression. Sharif et al. (2016) attack facial recognition systems by putting adversarial perturbations on glass frames. Their work demonstrates a successful physical attack under stable conditions and poses. Eykholt et al. (2018) proposes Robust Physical Perturbations (RP$_2$) in order to show that adding graffiti on a stop sign can cause it to be misclassified in both simulations and in the real world. Athalye et al. (2018) introduce Expectation over Transformation (EOT) and use it to print real-world objects which are adversarial given a range of physical and environmental conditions.

**Robust Adversarial Perturbations.** Li et al. (2019a) generates music which affects a voice assistant based system from picking up its wake word. Li et al. (2019b) presents a method for generating a targeted adversarial sticker which changes an image classifier's classification from one pre-specified class to another. Both of these methods rely on specific use cases and are tailored towards generating adversaries coming from strict distributions, e.g. (Li et al., 2019a) generates guitar music while (Li et al., 2019b) generates a small grid of dots. These works build on algorithms akin to our baseline approaches and are limited in scope to domain specific transformations. Our work provides a framework for improving robustness against a wide range of transformations in diverse domains and can be leveraged for improving the effectiveness of these attacks.

## 7 CONCLUSION

In this paper, we demonstrate that standard UAPs are highly susceptible to transformations, i.e. they fail to be universally adversarial under transformation. We propose a new method, `RobustUAP` to generate robust UAPs based upon obtaining probabilistic bounds on UAP robustness across an entire transformation space. Our experiments provide empirical evidence that this principled approach generates UAPs that are practically more robust under a wide range of transformation sets than those from the baseline methods.

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

APPENDIX

## A  SEMANTIC TRANSFORMATIONS

In this section, we discuss the semantic transformations used in the paper. Brightness and contrast can be represented via *bias* ($\beta$) and *gain* ($\alpha > 0$) parameters respectively. Formally, if $\mathbf{x}$ is the original image, then the transformed image, $\mathbf{x}'$, can be represented as

$$\mathbf{x}' = \alpha\mathbf{x} + \beta \tag{11}$$

Rotation, scaling, shearing, and translation are all affine transformations acting on the coordinate system, $c$, of the images instead of the pixel values, $\mathbf{x}$. In order to recover the pixel values and differentiate over the transformation, we will need sub-differentiable interpolation, see Appendix B. For finite dimensions, affine transformations can be represented as a linear coordinate map where the original coordinates are multiplied by an invertible augmented matrix and then translated with additional bias vector. Below, we give the general form for an affine transformation given augmented matrix $\mathbf{A}$, bias matrix $\mathbf{b}$, and input coordinates $c$. We can compute the output coordinates, $c'$, as

$$\begin{bmatrix} \mathbf{c}' \\ 1 \end{bmatrix} = \begin{bmatrix} [ccc|c] & \mathbf{A} & & \mathbf{b} \\ 0 & \dots & 0 & 1 \end{bmatrix} \begin{bmatrix} \mathbf{c} \\ 1 \end{bmatrix} \tag{12}$$

Below, we give the augmented matrix $\mathbf{A}$ and additional bias matrix $\mathbf{b}$ for rotation, scaling, shearing, and translation.

Rotation, $R(\theta)$, by $\theta$ degrees:

$$\mathbf{A} = \begin{pmatrix} \cos\theta & -\sin\theta \\ \sin\theta & \cos\theta \end{pmatrix}, \mathbf{b} = \begin{pmatrix} 0 \\ 0 \end{pmatrix} \tag{13}$$

Scaling, $Sc(p)$, by $p\%$:

$$\mathbf{A} = \begin{pmatrix} 1 + \frac{p}{100} & 0 \\ 0 & 1 + \frac{p}{100} \end{pmatrix}, \mathbf{b} = \begin{pmatrix} 0 \\ 0 \end{pmatrix} \tag{14}$$

Shearing, $Sh(m)$, by shear factor $m\%$:

$$\mathbf{A} = \begin{pmatrix} 1 & 1 + \frac{m}{100} \\ 0 & 1 \end{pmatrix}, \mathbf{b} = \begin{pmatrix} 0 \\ 0 \end{pmatrix} \tag{15}$$

Translation, $T(x, y)$, by $x$ pixels horizontally and $y$ pixels vertically:

$$\mathbf{A} = \begin{pmatrix} 0 & 0 \\ 0 & 0 \end{pmatrix}, \mathbf{b} = \begin{pmatrix} x \\ y \end{pmatrix} \tag{16}$$

## B  INTERPOLATION

Affine transformations may change a pixel's integer coordinates into non-integer coordinates. Interpolation is typically used to ensure that the resulting image can be represented on a lattice (integer) pixel grid. For this paper, we will be using bilinear interpolation, a common interpolation method which achieves a good trade-off between accuracy and efficiency in practice and is commonly used in literature (Xiao et al., 2018b; Balunović et al., 2019). Let $x_{i,j}$, $x'_{i,j}$ represent the pixel value at position $i, j$ for the original and transformed image respectively. Let $c'^x_{i,j}$, $c'^y_{i,j}$ represent the $x$-coordinate and $y$-coordinate of the pixel at $i, j$ after transformation. We define our transformed image by summing over all pixels $n, m \in [1 \dots H] \times [1 \dots W]$ where $H$ and $W$ represent the height and width of the image.

$$x'_{i,j} = \sum_{n}^{H} \sum_{m}^{W} x_{n,m} \max(0, 1 - |c'^x_{i,j} - m|) \max(0, 1 - |c'^y_{i,j} - n|) \tag{17}$$

This interpolation can be computed for each channel in the image. While interpolation is typically not differentiable, in order to generate adversarial examples using standard techniques we need a differentiable version of interpolation. (Jaderberg et al., 2015) introduces differentiable image sampling. Their method works for any interpolation method as long as the (sub-)gradients can be defined with respect to $x, c'_{i,j}$. For bilinear interpolation this becomes,

$$\frac{\partial x'_{i,j}}{\partial x_{n,m}} = \sum_n^H \sum_m^W \max(0, 1 - |c'^x_{i,j} - m|) \max(0, 1 - |c'^y_{i,j} - n|) \tag{18}$$

$$\frac{\partial x'_{i,j}}{\partial c'^x_{i,j}} = \sum_n^H \sum_m^W x_{n,m} \max(0, 1 - |c'^y_{i,j} - n|) \begin{cases} 1 & \text{if } m \geq |c'^x_{i,j} - m| \\ -1 & \text{if } m < |c'^x_{i,j} - m| \\ 0 & \text{otherwise} \end{cases} \tag{19}$$

## C  SGD ALGORITHM

Our SGD UAP algorithm is based on standard momentum based SGD while optimizing over the objective proposed in 5, the algorithm details can be seen in Algorithm 3.

---
**Algorithm 3** Stochastic Gradient Descent UAP Algorithm
---
1: Initialize $\mathbf{u_r} \leftarrow 0, \Delta\mathbf{u_r} \leftarrow 0$
2: **repeat**
3:    **for** $\mathbf{B} \in \mathbf{X}$ **do**
4:       Sample $\hat{t} \subset T$
5:       $\Delta\mathbf{u_r} \leftarrow \alpha\Delta\mathbf{u_r} - \frac{\nu}{|\hat{\mathbf{x}}| \times |\hat{t}|} \sum_{i=1}^{|\hat{\mathbf{x}}|} \sum_{j=1}^{|\hat{t}|} \nabla L[f(\hat{\mathbf{x}_i} + \hat{t}_j(\mathbf{u_r})), f(\hat{\mathbf{x}_i})]$
6:       Update the perturbation with projection:
7:       $\mathbf{u} \leftarrow \mathcal{P}_{p,\epsilon}(\mathbf{u_r} + \Delta\mathbf{u_r})$
8:    **end for**
9: **until** $ASR_R(f, \mathbf{X}, T, \gamma, \mathbf{u_r}) < \zeta$
---

## D  ITERATIVE UAP ALGORITHM

Moosavi-Dezfooli et al. (2017) introduces an iterative UAP algorithm, the algorithm can be seen in Algorithm 4.

---
**Algorithm 4** Iterative Universal Perturbation Algorithm (Moosavi-Dezfooli et al. (2017))
---
1: Initialize $\mathbf{u} \leftarrow 0$
2: **repeat**
3:    **for** $\mathbf{x_i} \in \mathbf{X}$ **do**
4:       **if** $\hat{f}(\mathbf{x_i} + \mathbf{u}) = \hat{f}(\mathbf{x_i})$ **then**
5:          Compute minimal adversarial perturbation:
6:          $\Delta\mathbf{u} \leftarrow \arg\min_{\mathbf{r}} ||\mathbf{r}||_2 \text{ s.t. } \hat{f}(\mathbf{x_i} + \mathbf{u} + \mathbf{r}) \neq \hat{f}(\mathbf{x_i})$
7:          Update the perturbation with projection:
8:          $\mathbf{u} \leftarrow \mathcal{P}_{p,\epsilon}(\mathbf{u} + \Delta\mathbf{u})$
9:       **end if**
10:   **end for**
11: **until** $ASR_U(f, \mathbf{X}, \mathbf{u}) < \gamma$
---

## E  EXPERIMENT PARAMETERS

In our experiments, we have capped all algorithms at 5 epochs or if they have achieved an $ASR_R$ of 0.95. The UAPs are trained with the same transformation set that they are evaluated on. For algorithms running PGD internally, we have capped the number of iterations to 40.

## F    FURTHER EVALUATION OF UNIFORM NOISE

Results in Table 2.

| DATASET | TRANSFORMATION SET | STANDARD UAP | SGD | STANDARD UAP_RP | ROBUST UAP |
|---------|--------------------|--------------|-----|-----------------|------------|
| ILSVRC 2012 | $U(0.1)$ | 81.6% | 94.2% | 91.3% | **99.0%** |
|             | $U(0.3)$ | 10.7% | 68.9% | 42.7% | **96.1%** |
| CIFAR-10 | $U(0.1)$ | 66.0% | 98.1% | 96.1% | **100%** |
|          | $U(0.3)$ | 5.8% | 96.1% | 47.6% | **100%** |

Table 2: Robust ASR with uniform random noise, $\gamma = 0.8$.

## G    UAP PERFORMANCE AGAINST SEMANTIC TRANSFORMATIONS ON CIFAR-10

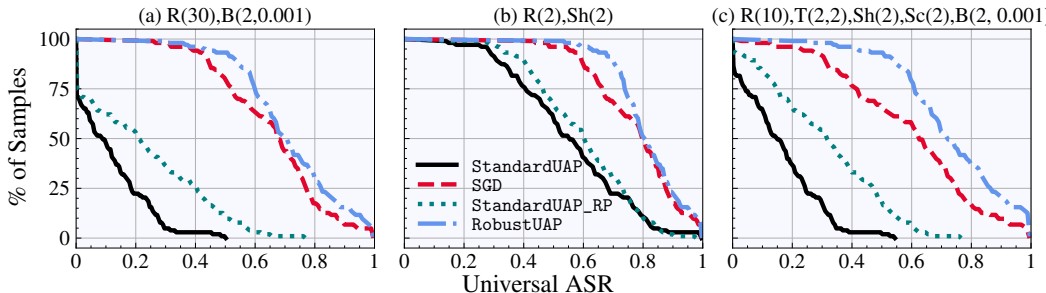

Figure 6: For each method, a point $(x, y)$ in the corresponding line represents the percentage of sampled UAPs ($y\%$) with Universal ASR $> x$ for the different semantic transformations on CIFAR-10.

## H    AVERAGE $\text{ASR}_U$ AND $\text{ASR}_R$ WITH DIFFERENT $\gamma$'S

We provide additional metrics computed on the same set of transformations, datasets, and models as in Table 1. In Table 3, we present the Average $\text{ASR}_U$ rather than $\text{ASR}_R$. The average shows us that our RobustUAP algorithm creates UAPs which after transformation on average are better UAPs than all other algorithms. We observe that the average shows us that even standard UAPs aren't completely ineffective after transformation they just have a very low chance of being highly effective.

| DATASET | TRANSFORMATION SET | STANDARD UAP | SGD | STANDARD UAP_RP | ROBUST UAP |
|---------|--------------------|--------------|-----|-----------------|------------|
| ILSVRC 2012 | $R(20)$ | 16.3% | 71.5% | 24.7% | **81.3%** |
|             | $T(2,2)$ | 52.6% | 82.6% | 55.4% | **85.4%** |
|             | $Sc(5), R(5), B(5, 0.01)$ | 44.9% | 76.3% | 58.5% | **82.2%** |
|             | $R(10), T(2,2), Sh(2), Sc(2), B(2, 0.001)$ | 13.6% | 64.8% | 29.0% | **75.3%** |
| CIFAR-10 | $R(30), B(2, 0.001)$ | 9.9% | 66.8% | 22.2% | **73.4%** |
|          | $R(2), Sh(2)$ | 57.1% | 78.8% | 61.2% | **82.9%** |
|          | $R(10), T(2,2), Sh(2), Sc(2), B(2, 0.001)$ | 16.2% | 61.2% | 32.6% | **76.4%** |

Table 3: Average Universal ASR of our Robust UAP algorithms and the standard UAP (Moosavi-Dezfooli et al., 2017) method.

In Table 4, we present $\text{ASR}_R$ computed at $\gamma = [0.5, 0.7]$ rather than $\gamma = 0.6$. This table shows a similar story to above, and shows that our algorithm produces better results under a variety of success thresholds.

| Dataset | Transformation Set | Standard UAP 0.5 | 0.7 | SGD 0.5 | 0.7 | Standard UAP_RP 0.5 | 0.7 | Robust UAP 0.5 | 0.7 |
|---|---|---|---|---|---|---|---|---|---|
| ILSVRC 2012 | $R(20)$ | 1.9% | 0.0% | 88.3% | 58.3% | 10.7% | 1.0% | **98.1%** | **76.7%** |
| | $T(2,2)$ | 51.5% | 21.4% | **100%** | 84.5% | 57.3% | 23.3% | **100%** | **91.3%** |
| | $Sc(5), R(5), B(5,0.01)$ | 38.8% | 11.7% | 96.1% | 67.0% | 64.1% | 25.2% | **99.0%** | **87.4%** |
| | $R(10), T(2,2), Sh(2), Sc(2), B(2,0.001)$ | 1.9% | 0.0% | 82.5% | 38.8% | 12.6% | 1.0% | **95.1%** | **59.2%** |
| CIFAR-10 | $R(30), B(2,0.001)$ | 1.0% | 0.0% | 80.6% | 43.7% | 12.6% | 1.0% | **93.2%** | **49.5%** |
| | $R(2), Sh(2)$ | 62.1% | 22.3% | 96.1% | 68.9% | 68.0% | 30.1% | **99.0%** | **89.3%** |
| | $R(10), T(2,2), Sh(2), Sc(2), B(2,0.001)$ | 2.9% | 0.0% | 67.0% | 38.8% | 19.4% | 1.0% | **93.2%** | **55.3%** |

Table 4: Robust ASR of our Robust UAP algorithms and the standard UAP (Moosavi-Dezfooli et al., 2017) method with $\gamma = [0.5, 0.7]$.

| Dataset | StandardUAP | SGD | StandardUAP_RP | RobustUAP |
|---|---|---|---|---|
| ILSVRC 2012 | **95.5%** | 85.6% | 82.3% | 91.3% |
| CIFAR-10 | **96.2%** | 89.3% | 84.0% | 93.7% |

Table 5: Universal ASR of our Robust UAP algorithms and the standard UAP method.

# I  COMPARISON ON NON-ROBUST UNIVERSAL ASR METRIC

We compare our robust UAPs to standard UAPs on the non-robust universal ASR metric, see Table 5. All robust UAPs are generated to be robust against $R(10), T(2,2), Sh(2), Sc(2), B(2,0.001)$. We observe that at the same $l_2$-norm all robust UAPs achieve a lower universal ASR than the standard UAP algorithm. This result is not too surprising as solving the optimization problem for robust UAP is significantly more difficult. We further observe that our RobustUAP algorithm is the most effective in comparison to the other robust baseline approaches.

# J  ADDITIONAL MODELS

We also provide additional data on our methods evaluated on the same transformations and datasets but on different models. In this case, we use ResNet-18 (He et al., 2015) for CIFAR-10 and MobileNet (Howard et al., 2017) for ILSVRC 2012. Results can be seen in Table 6. We observe similar performance across models suggesting that the performance of the attacks is more directly tied to transformation set and dataset.

| Dataset | Model | Transformation Set | Standard UAP | SGD | Standard UAP_RP | Robust UAP |
|---|---|---|---|---|---|---|
| ILSVRC 2012 | MobileNet | $R(20)$ | 8.1% | 71.2% | 2.6% | **85.0%** |
| | | $T(2,2)$ | 40.9% | 98.7% | 54.3% | **99.6%** |
| | | $Sc(5), R(5), B(5,0.01)$ | 16.3% | 94.5% | 44.3% | **96.3%** |
| | | $R(10), T(2,2), Sh(2), Sc(2), B(2,0.001)$ | 4.1% | 75.7% | 8.6% | **86.2%** |
| CIFAR-10 | ResNet-18 | $R(30), B(2,0.001)$ | 0.9% | 67.8% | 6.4% | **74.9%** |
| | | $R(2), Sh(2)$ | 49.9% | 99.5% | 49.1% | **99.8%** |
| | | $R(10), T(2,2), Sh(2), Sc(2), B(2,0.001)$ | 8.0% | 70.8% | 12.2% | **83.8%** |

Table 6: Robust ASR on Resnet-18 for CIFAR-10 and MobileNet for ILSVRC 2012.

# K  COMMON CORRUPTIONS

We also evaluate robust UAP against the 2D fog transformations in (Kar et al., 2022). We set the shift intensity of the fog to be 1 and train our robust UAPs to be robust against random fog perturbations. We observe similar results to the transformations we experiment with above. The graph of the results can be seen in Figure 7.

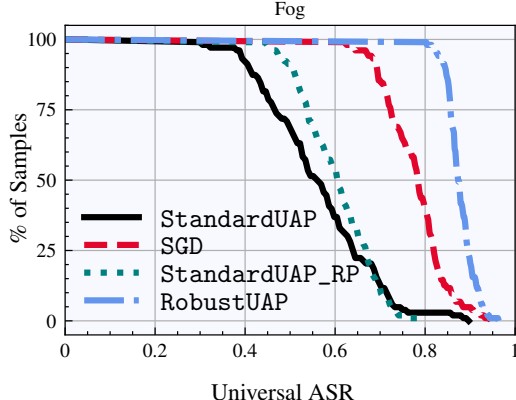

Figure 7: For each method, a point $(x, y)$ in the corresponding line represents the percentage of sampled UAPs ($y\%$) with Universal ASR $> x$ for the different semantic transformations on ILSVRC-2012.

## L    ALGORITHM RUNTIMES

We compare the average runtimes of the different methods on one of our most challenging $R(10), T(2, 2), Sh(2), Sc(2), B(2, 0.001)$ transformation set on ILSVRC-2012 and $n = 738$. The results are in Table 7. We observe that RobustUAP is the slowest algorithm and SGD is the fastest. RobustUAP uses EstimateRobustness in each loop and thus with high $n$ it requires much more time to compute. The extra computation enables Robust UAP to obtain better robustness than all baselines. On the same set of transformations and dataset we observe that one iteration of EstimateRobustness on the entire test set takes on average 19 minutes. When running EstimateRobustness in the RobustUAP loop, each call takes 36 seconds for a batch size of 32.

| ALGORITHM | TIME(MIN) |
|---|---|
| STANDARD UAP | 37 |
| SGD | 32 |
| STANDARD UAP_RP | 43 |
| ROBUST UAP | 118 |

Table 7: Average Runtime for Robust UAP algorithms

## M    EFFECT OF COMPUTE TIME ON ROBUSTNESS

Previous sections highlight SGD as the most competitive algorithm to RobustUAP in terms of performance. However, in the previous section we note that SGD takes significantly less time to run. In this section, we investigate how RobustUAP performs with limited compute time as well as how SGD performs with increased runtime. We first add results to the ILSVRC 2012 part of Table 1 by also computing RobustUAP performance when limited to the same amount of time that SGD takes. Table 8 shows that RobustUAP outperforms SGD even when its compute time is limited with up to 9% more robustness on our most challenging transformation $R(10), T(2, 2), Sh(2), Sc(2), B(2, 0.001)$.

| DATASET | TRANSFORMATION SET | SGD | ROBUST UAP | RESTRICTED ROBUST UAP |
|---|---|---|---|---|
| ILSVRC 2012 | $R(20)$ | 69.9% | **93.2%** | 72.9% |
| | $T(2, 2)$ | 96.1% | **97.1%** | 96.9% |
| | $Sc(5), R(5), B(5, 0.01)$ | 85.4% | **96.1%** | 86.3% |
| | $R(10), T(2, 2), Sh(2), Sc(2), B(2, 0.001)$ | 63.1% | **86.4%** | 72.0% |

Table 8: Robust ASR of RobustUAP restricted to the same amount of compute time as SGD.

Next, we vary the number of SGD iterations. We compute the robust ASR on ILSVRC for robustness against $R(10), T(2,2), Sh(2), Sc(2), B(2, 0.001)$. Figure 8, shows the robust ASR achieved by SGD over time, here we observe that SGD's performance flatlines after a small number of iterations and seems to be unable to surpass about 65. Here `SGD` is allowed to continue to run past where it would usually stop (at around 250 iterations), in this experiment we allow it to go to 1250 iterations which is about the same amount of time that `RobustUAP` takes to run. `RobustUAP` is able to achieve a performance of 72 even when restricted to the amount of compute time of base `SGD` (It achieves 86.4 when unrestricted). These two results in combination show that `RobustUAP` is able to find more robust UAPs than `SGD` whose performance stabilizes.

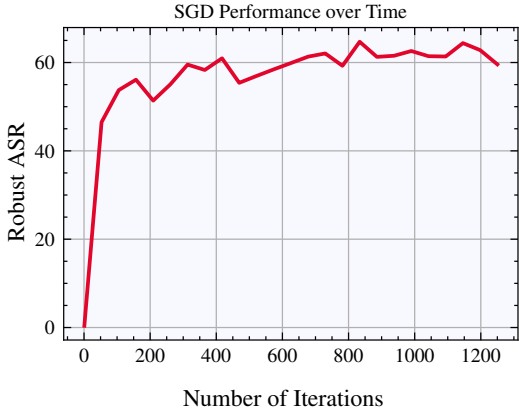

Figure 8: The Robust ASR with $\gamma = 0.6$ for `SGD` over time

## N    ROBUSTNESS ON HOLD-OUT TRANSFORMATIONS

In this section, we measure the effectiveness of our robust UAPs against hold-out transformations. In this experiment, we learn a UAP which is robust to $R(5)$ and obtain a robust ASR of 96.2 at $\gamma = 0.6$. We then measure its effectiveness against $Sc(5)$ and get a robust ASR of 85.4, in contrast, a robust UAP trained directly to be robust to $Sc(5)$ obtains robust ASR of 98.1. Next, we measure the robustness of UAP trained against $R(5)$ when subjected to transformations from $B(5, 0.01)$. Here we get a robust ASR of 97.3, whereas a robust UAP trained to be robust to $B(5, 0.01)$ obtains a robust ASR of 99.2. Finally, we test the robust UAP on $R(5), Sc(5), B(5, 0.01)$ and get a robust ASR of 83.1. Our previous results show that a UAP trained to be robust against these parameters directly can obtain a robust ASR of 96.1. In each case, our UAP maintains robustness on hold-out transformations but has lower performance compared to robust UAPs trained directly to be robust to those transformations.

## O    TARGETED ATTACK

So far in this paper we have focused on untargeted attacks, i.e. attacks which aim to degrade the general performance of the model. Targeted attacks are also possible with both standard adversarial attack methods and universal adversarial perturbation methods. Here, we can simply turn our algorithm from untargeted to targeted by replacing the loss function. We would like to have target class, A, be classified as target class, B. Instead of maximizing the expected value of the cross entropy loss we can instead formulate the loss based on maximizing B while minimizing A similar to (Benz et al., 2020). For ILSVRC 2012, we randomly select a couple of target classes and perform this attack, for each of these cases, we train our robust UAP to be robust to $R(10), T(2,2), Sh(2), Sc(2), B(2, 0.001)$. Table 9 shows our results for robust ASR with $\gamma = 0.6$. We are measuring our robust ASR of turning class A into class B and observe similar results with `RobustUAP` being the most robust followed by `SGD`. It is also interesting to note that different random combinations lead to more or less success, i.e. it is easier to turn a dog into another dog than perfume into a padlock.

| Dataset | Target Class | Standard UAP | SGD | Standard UAP_RP | Robust UAP |
|---|---|---|---|---|---|
| ILSVRC-2012 | TOY POODLE → MALTESE DOG | 42.4% | 99.1% | 85.6% | **99.8**% |
| | PERFUME → PADLOCK | 0.0% | 63.8% | 5.1% | **76.4**% |

Table 9: Robust ASR of `RobustUAP` for target to target attack compared to the three baselines with $\gamma = 0.6$.

## P  DATA EFFICIENCY

In this section, we will evaluate the data efficiency of `RobustUAP`. We use `RobustUAP` to generate UAPs robust to $R(10), T(2, 2), Sh(2), Sc(2), B(2, 0.001)$ on ILSVRC-2012 with differing amounts of training data. The results can be seen in Figure 9. These results show that the algorithm is able to achieve good performance at 500 data points but continues to improve up to 4000 data points. After that it seems to stagnate.

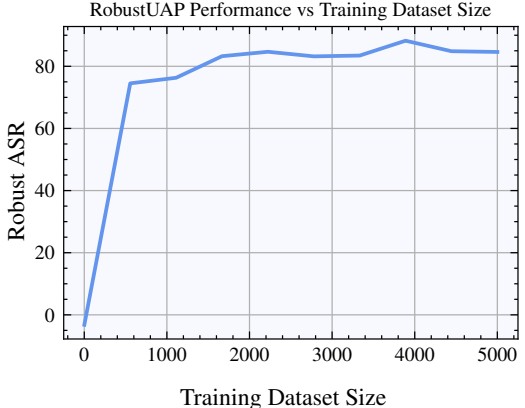

Figure 9: Robust ASR with $\gamma = 0.6$ for `RobustUAP` with differing amounts of training data

## Q  TRANSFERABILITY

In this section, we will evaluate the transferability of `RobustUAP`. Previous works on UAPs (Moosavi-Dezfooli et al., 2017) show that UAPs are transferable across different models. Here, we will evaluate whether robust UAPs exhibit the same behavior for robustness. The robust UAPs studied here are generated with `RobustUAP` on $R(10), T(2, 2), Sh(2), Sc(2), B(2, 0.001)$ for ILSVRC-2012 with $\gamma = 0.6$. We use a variety of models: Inception-v3 (Szegedy et al., 2016), MobileNet (Howard et al., 2017), Inception-v3 trained to be robust on $R(20)$ (InceptionR20), Inception-v3 trained to be robust on horizontal flips (InceptionHF), and ViT (Dosovitskiy et al., 2020). Table 10 shows us that our robust UAPs are transferable between different architectures. Our results show that robust UAPs transfer their robustness properties between architectures and models. Ignoring ViT, on all of the Inception and MobileNet models, the generated UAPs maintain at least 65% robust ASR when transferred to each other. This transfer is less but still significant for ViT where it maintains at least 32% robustness when transferred to or from the other models.

| | TARGET MODEL | | | | |
|---|---|---|---|---|---|
| SOURCE MODEL | INCEPTION | MOBILENET | INCEPTIONR20 | INCEPTIONHF | VIT |
| INCEPTION | **86.4**% | 65.2% | 75.2% | 78.5% | 35.1% |
| MOBILENET | 74.3% | **86.2**% | 67.3% | 68.6% | 38.3% |
| INCEPTIONR20 | 80.1% | 67.3% | **81.3**% | 73.1% | 32.0% |
| INCEPTIONHF | 77.8% | 70.9% | 75.8% | **83.8**% | 34.6% |
| VIT | 41.2% | 32.4% | 43.2% | 39.7% | **88.5**% |

Table 10: Robust ASR when UAP is learned on source model and transfered to target model.

# R TRANSFORMER-BASED MODELS

Recently, transformers have become popular as a new architecture for deep learning models for computer vision tasks. In this section, we evaluate the effectiveness of robust UAPs against one such model, ViT (Dosovitskiy et al., 2020). Benz et al. (2021) has shown that standard UAPs are still effective against transformer based architectures. In Table 11 we can see that we get similar results compared to our results on Inception and MobileNet. This shows that our methods work against transformer based models as well.

| DATASET | MODEL | TRANSFORMATION SET | STANDARD UAP | SGD | STANDARD UAP_RP | ROBUST UAP |
|---------|-------|---------------------|--------------|-----|------------------|------------|
| ILSVRC-2012 | ViT | $R(10), T(2,2), Sh(2), Sc(2), B(2, 0.001)$ | 2.0% | 72.1% | 12.9% | **88.5%** |

Table 11: Robust ASR of `RobustUAP` compared to the three baselines for ViT.

# S ROBUST UAPs AGAINST ROBUSTLY TRAINED NETWORKS

In this section, we are interested in seeing whether training networks to be robust against the same transformations that the UAP is trying to be robust against is helpful. For this, we trained two new Inception-v3 networks. Because of time limitations, we started with our base Inception-v3 network and fine-tuned it using data augmentations. For the first network InceptionR20, we augmented the data by adding random rotations within 20 degrees. For the second network InceptionHF, we augmented the data by adding horizontal flips. We then crafted UAPs robust against rotations and flips on InceptionR20 and InceptionHF respectively. The results can be seen in Table 12. We can compare the $R(20)$ results to those from our normal inception network. We postulate that since the network has received some additional robustness training it is harder to attack, and thus we should see slightly lower robustness scores. However, it seems that training the network to be robust to $R(20)$ does not significantly effect the ability to create robust UAPs. The horizontal flips seems like it might be too easy of a transformation as even standard UAP performs quite well for robust ASR.

| DATASET | MODEL | TRANSFORMATION SET | STANDARD UAP | SGD | STANDARD UAP_RP | ROBUST UAP |
|---------|-------|---------------------|--------------|-----|------------------|------------|
| ILSVRC-2012 | INCEPTIONR20 | $R(20)$ | 6.3% | 72.4% | 10.2% | **81.3%** |
| | INCEPTIONHF | $HF$ | 81.3% | 99.5% | 89.7% | **99.6%** |

Table 12: Robust ASR of `RobustUAP` compared to the three baselines for robust networks.

# T ABLATION ON OPTIMIZATION STRATEGY

In this section, we study the effect of using different optimizers in addition to SGD. We use a variety of standard PyTorch optimizers, Adam, Adamax, Adagrad, and RMSProp. We formulate the optimization problem in the same way but instead use these algorithms in order to optimize our perturbation. We compute these results on ILSVRC-2012 with Inception-v3 and use $R(10), T(2,2), Sh(2), Sc(2), B(2, 0.001)$ as the transformation set and with $\gamma = 0.6$. The results can be seen in Table 13. We see that the optimization strategy has some affect on the results and that SGD performs the best. We also found that SGD performed marginally faster than the rest of the approaches.

| OPTIMIZER | $ASR_R$ |
|-----------|---------|
| SGD | **63.1**% |
| ADAM | 59.7% |
| ADAMAX | 60.1% |
| ADAGRAD | 62.3% |
| RMSPROP | 58.3% |

Table 13: Comparison of different optimization strategies.

