# OpenReview forum: "Robust Universal Adversarial Perturbations"
_ICLR.cc/2023/Conference — Submitted to ICLR 2023_

### Official Review · Reviewer_pw3U · 2022-10-13

**Confidence:** 3
**Correctness:** 4
**Technical Novelty And Significance:** 2
**Empirical Novelty And Significance:** 3
**Recommendation:** 6

**Clarity, Quality, Novelty And Reproducibility:**

The method may be novel, although I am not aware of other recent work on UAPs.  The writing is good in terms of the grammar, but the simple and intuitive method is hidden behind a wall of theory.

**Strength And Weaknesses:**

How does the ability to craft perturbations which are robust under transformations depend on the invariance properties of the victim network to such transformations?  For example, is it easier to craft UAPs which are robust to horizontal flips for models trained with horizontal flip data augmentations.  How does this depend on the model used for crafting as well?

This paper contains virtually no ablations, for example on the choice of optimizer in the attack.  The authors use a different optimization strategy for example in the SGD baseline instead of PGD, so that is important to ablate.

My biggest qualm with this work is that the paper is packed with theory and notation, yet the proposed method is extremely simple and does not require any of the theory for motivation.  In fact, while I was reading the theory, I wrote down a note to myself asking if the authors tried a very simple and obvious baseline, and that baseline turned out to be exactly their proposed method.  The technique of EOT is widely used in the adversarial example and poisoning literature, for example, so it makes sense to use it here as well.  Simplicity is a good thing as simple methods are easier to understand and implement, but loading up a paper with entirely unnecessary theory and notation so that it is harder for the reader to understand the work defeats the purpose.  The algorithm only appears on page 6 of 9 in a method paper.

**Summary Of The Paper:**

The paper presents a simple improvement to universal adversarial perturbations to make them robust to transformations.  The paper motivates empirical experiments via theoretical observations.

**Summary Of The Review:**

I like the simplicity of the approach and the empirical robustness gains, but the presentation is hard to follow, and there are numerous experiments missing that would have been a better use of space than the theory.

---

> ### Author Response · Authors · 2022-11-14
> **Response to Reviewer pw3U**
>
> We thank the reviewer for their constructive feedback. We are happy that the reviewer liked our approach and the results that it obtained. Below, we address the concerns of the reviewer and detail our experiments addressing the additional experiments requested by the reviewer. We would be happy to perform any additional experiments requested by the reviewer.
>
> >**Q1: How does the ability to craft perturbations which are robust under transformations depend on the invariance properties of the victim network to such transformations? How does this depend on the model used for crafting as well?**
>
> **R1:** We investigate this question by using data augmentation to train networks robust to simple transformations. For this study, we trained a Inception-v3 network robust to R(20), InceptionR20, and one robust to horizontal flips, InceptionHF. We then use RobustUAP to craft UAPs robust to R(20) and horizontal flips for InceptionR20 and InceptionHF respectively. The results can be seen in Appendix S and below. These results show that the ability to craft perturbations which are robust under transformations do not significantly depend on the invariance properties of the victim network to such transformations.
>
> | Model        | Transformation Set | Standard UAP | SGD   | Standard UAP_RP | RobustUAP |
> |--------------|--------------------|--------------|-------|-----------------|-----------|
> | InceptionR20 | R(20)              | 6.3%         | 72.4% | 10.2%           | **81.3%** |
> | InceptionHF  | HF                 | 81.3%        | 99.5% | 89.7%           | **99.6%** |
>
>
> We further investigate how this depends on the model used for crafting by performing a transferability study (also see **R5** for jAxg), see Appendix Q or below. We find that the similar model architectures that have been trained with different robustness properties seem to transfer robust UAPs quite well. This suggests that the model used for crafting does not significantly affect the ability to craft these perturbations either.
>
> |              |           | Target    | Model        |             |           |
> |--------------|-----------|-----------|--------------|-------------|-----------|
> | Source Model | Inception | MobileNet | InceptionR20 | InceptionHF | ViT       |
> | Inception    | **86.4%** | 65.2%     | 75.2%        | 78.5%       | 35.1%     |
> | MobileNet    | 74.3%     | **86.2%** | 67.3%        | 68.6%       | 38.3%     |
> | InceptionR20 | 80.1%     | 67.3%     | **81.3%**    | 73.1%       | 32.0%     |
> | InceptionHF  | 77.8%     | 70.9%     | 75.8%        | **83.8%**   | 34.6%     |
> | ViT          | 41.2%     | 32.4%     | 43.2%        | 39.7%       | **88.5%** |
>
> >**Q2: This paper contains virtually no ablations, for example on the choice of optimizer in the attack. The authors use a different optimization strategy for example in the SGD baseline instead of PGD, so that is important to ablate.**
>
> **R2:** We study the effect of using different optimizers (compared to SGD) by using a variety of standard PyTorch optimizers, Adam, Adamax, Adagrad, and RMSProp. We formulate the optimization problem in the same way but instead use these algorithms in order to optimize our perturbation. The results can be seen in Appendix T or below. We see that the optimization strategy has some effect on the results and that SGD performs the best. We also found that SGD performed marginally faster than the rest of the approaches.
>
> | Optimizer | $ASR_R$   |
> |-----------|-----------|
> | SGD       | **63.1%** |
> | Adam      | 59.7%     |
> | Adamax    | 60.1%     |
> | Adagrad   | 62.3%     |
> | RMSProp   | 58.3%     |
>
> >**Q3: Simplicity is a good thing as simple methods are easier to understand and implement, but loading up a paper with entirely unnecessary theory and notation so that it is harder for the reader to understand the work defeats the purpose. The algorithm only appears on page 6 of 9 in a method paper.**
>
> **R3:** We believe that our theory is important for this work as it guides the design of our algorithm. The simple and obvious baselines (the SGD approach and the modification on the standard UAP algorithm) do not achieve adequate robustness as they do not explicitly optimize for the robustness of their generated UAPs over the considered perturbation region. Our final RobustUAP algorithm tries to solve our robust UAP formulation from eq. (5) by leveraging a sampling strategy trying to approximate the expectation in eq. (5) as close as possible. This helps our method in achieving empirically better results than the baselines. Our theory also allows us to give probabilistic bounds on the robustness of our attack.
>
> We have additionally provided many more experiments in the appendix as suggested by other reviewers and hope that these fill in some gaps as pointed out by this reviewer.

---

> > ### Comment · Reviewer_pw3U · 2022-11-25
> > **Thanks for your response**
> >
> > I have increased my score since your rebuttal as well as additional experiments run for other reviewers address most of my questions.  I still think the paper is tougher than it needs to be to read because of the wall of theory in front of a very simple method.

---

> > > ### Author Response · Authors · 2022-11-30
> > > **Thank You**
> > >
> > > We thank the reviewer for their response and revised score, we will continue to work on the clarity of the paper.

---

### Official Review · Reviewer_jAxg · 2022-10-19

**Confidence:** 4
**Correctness:** 3
**Technical Novelty And Significance:** 2
**Empirical Novelty And Significance:** 2
**Recommendation:** 3

**Clarity, Quality, Novelty And Reproducibility:**

## Clarity
This work clearly written and technically sound to the best of my judgment.

## Quality
The paper quality is sufficient in my opinion.

## Novelty
In my judgment, this work holds only limited novelty.

## Reproducibility
This work is reproducible (code provided) to the best of my judgment.


**Strength And Weaknesses:**

## Strengths
(+) This work deals with the topic of the robustness of Universal Adversarial Perturbations (UAPs). This is an important topic, since, as the authors demonstrate, the effectiveness of UAPs decreases under image transformations.
(+) The provided experimental results are convincing, showing that the proposed method outperforms the compared baselines.

## Weaknesses
(-) The contribution of this work is limited. In essence, this work shows that to overcome the susceptibility of UAPs to transformations, transformations need to be incorporated into the UAP crafting process. The increased robustness is hence expected.
(-) The proposed algorithm has only limited novelty and is a collection of proven methods in the literature, namely batch-training [A, B], expectation over transformations [Athalye et al., 2018], and PGD [Madry et al., 2017]. It would be beneficial if the authors could differentiate their algorithm from the ones in the literature [Shafahi et al., 2020, A, B].
(-) To further demonstrate the robustness of the UAP, the authors should evaluate the robust UAPs on hold-out transformation, which were not observed during training.
(-) The authors mainly evaluated the robust UAP in the context of untargeted attack success rate. It would be further interesting to see a performance evaluation of the targeted attack success rate.
(-) I am further curious about the data efficiency and transferability of robust UAPs.
* Data efficiency: How do robust UAPs perform when only limited data is accessible during the training process? Please also note, that in the literature data-free methods exist.
* Transferability: It is common to analyze the transferability of UAPs [Moosavi-Dezfooli et al. 2017, A, B, C]. I would be curious about the transferability of robust UAPs, compared to existing methods.

(-) The authors mainly evaluated robust UAPs on VGG, Inception, and ResNet18. Recently, transformer architectures gained popularity. I am curious if the concept of robust UAPs holds for transformer-based models as well.
(-) The related work section is insufficient. To name only a few missing works: [A-E].

[A] Generalizable Data-free Objective for Crafting Universal Adversarial Perturbations; T-PAMI 2018
[B] Understanding Adversarial Examples from the Mutual Influence of Images and Perturbations; CVPR 2020
[C] Regional Homogeneity: Towards Learning Transferable Universal Adversarial Perturbations Against Defenses; ECCV 2020
[D] Art of singular vectors and universal adversarial perturbations; CVPR 2018
[E] Defense against Universal Adversarial Perturbations; CVPR 2018


**Summary Of The Paper:**

This work introduces robust Universal Adversarial Perturbations (UAPs). The objective of robust UAPs is to increase the resilience of UAPs against image transformations. Robust UAPs are obtained by incorporating transformation functions into the UAP generation process. Evaluation on CIFAR10 and ImageNet demonstrate the increased performance of robust UAPs compared to baseline approaches.

**Summary Of The Review:**

While it is indeed true that the efficiency of UAPs suffers under transformations, simply showing that this can be mitigated by introducing transformations into the crafting process does in my opinion not hold enough contribution to be accepted at ICLR. Additionally, as pointed out in my weaknesses section, this work lacks several crucial evaluations to fully judge the effectiveness of the proposed robust UAPs.

---

> ### Author Response · Authors · 2022-11-14
> **Response to Reviewer jAxg (3/3)**
>
> [A] Generalizable Data-free Objective for Crafting Universal Adversarial Perturbations; T-PAMI 2018
>
> [B] Understanding Adversarial Examples from the Mutual Influence of Images and Perturbations; CVPR 2020
>
> [C] Regional Homogeneity: Towards Learning Transferable Universal Adversarial Perturbations Against Defenses; ECCV 2020
>
> [D] Art of singular vectors and universal adversarial perturbations; CVPR 2018
>
> [E] Defense against Universal Adversarial Perturbations; CVPR 2018
>
> [F] Double targeted universal adversarial perturbations; ACCV 2020
>
> [G] An image is worth 16x16 words: Transformers for image recognition at scale; ICLR 2021
>
> [H] Universal Adversarial Perturbations; CVPR 2017

---

> ### Author Response · Authors · 2022-11-14
> **Response to Reviewer jAxg (2/3)**
>
> >**Q5: Transferability: It is common to analyze the transferability of UAPs [H, A, B, C]. I would be curious about the transferability of robust UAPs, compared to existing methods.**
>
> **R5:** In order to measure the transferability of robust UAPs, we take the UAPs generated for ILSVRC from different models. Here we use, Inception-v3, MobileNet, Inception-v3 trained to be robust on R(20) (InceptionR20), Inception-v3 trained to be robust on Horizontal Flips (InceptionHF), and ViT. We report the Robust ASR in the cross-table seen in Appendix Q and below. Our results show that robust UAPs transfer their robustness properties between architectures and models. Ignoring ViT, on all of the Inception and MobileNet models, the generated UAPs maintain at least 65\% robust ASR when transferred to each other. This transfer is less but still significant for ViT where it maintains at least 32\% robustness when transferred to or from the other models.
>
>
> |              |           | Target    | Model        |             |           |
> |--------------|-----------|-----------|--------------|-------------|-----------|
> | Source Model | Inception | MobileNet | InceptionR20 | InceptionHF | ViT       |
> | Inception    | **86.4%** | 65.2%     | 75.2%        | 78.5%       | 35.1%     |
> | MobileNet    | 74.3%     | **86.2%** | 67.3%        | 68.6%       | 38.3%     |
> | InceptionR20 | 80.1%     | 67.3%     | **81.3%**    | 73.1%       | 32.0%     |
> | InceptionHF  | 77.8%     | 70.9%     | 75.8%        | **83.8%**   | 34.6%     |
> | ViT          | 41.2%     | 32.4%     | 43.2%        | 39.7%       | **88.5%** |
>
>
> >**Q6: The authors mainly evaluated robust UAPs on VGG, Inception, and ResNet18. Recently, transformer architectures gained popularity. I am curious if the concept of robust UAPs holds for transformer-based models as well.**
>
> **R6:** UAP effectiveness has been explored for transformer-based models. We implement and train ViT [G] and show our results in Appendix R and below. Our result shows that our UAPs generated with our method are robust on transformer-based models as well.
>
>
> | Model | Standard UAP | SGD   | Standard UAP_RP | RobustUAP |
> |-------|--------------|-------|-----------------|-----------|
> | ViT   | 2.0%         | 72.1% | 12.9%           | **88.5%** |
>
>
> >**Q7: The related work section is insufficient. To name only a few missing works: [A-E]**
>
> **R7:** We have added comparisons to [A-E] in related work. [A] GD-UAP is a data-free approach to learning UAPs. Their method works by trying to create spurious activations in the neural network layers. [B] This paper explores the connection between universal perturbations and dominant features. Using their insight they generate a method for creating targeted UAPs using a proxy dataset. Their method involves a specially crafted loss function which is then optimized for. [C] This paper devises a gradient transformer module which they attach to any iterative gradient based adversarial attack method in order to induce regionally homogeneous perturbations. While their method is not inherently a UAP method they note that by picking the right parameters their method becomes input independent. [D] This paper notes that by computing the (p,q) singular vectors of the jacobian matrices of hidden layers in a network they can get UAPs with a very small amount of data (~64). They propose a stochastic power method in order to solve for these singular vectors. [E] This paper proposes a perturbation rectifying network (PRN) in order to defend against UAPs. They train their PRN by feeding it UAPs during training, they generate these UAPs by creating an efficient ‘synthetic’ perturbation generator. Like the previous papers this one does not consider robustness of the perturbation itself but does consider robustness against universal perturbations.
>
> All of these papers propose methods for UAP generation, but they do not consider the robustness of their generated UAPs and do not provide any probabilistic bounds on the robustness of their generated UAPs. We have shown above (in **R4**) that [A] fairs similarly to standard UAP in that it is not robust. We can further experiment with the remaining methods, but our experimental results suggest that robustness to these transformations does not come for free and one must explicitly optimize for them.
>
> We have provided more experiments in the appendix as recommended by other reviewers that hopefully provide additional results. We will be happy to provide additional results as needed by the reviewer.

---

> ### Author Response · Authors · 2022-11-14
> **Response to Reviewer jAxg (1/3)**
>
> We thank the reviewer for their constructive feedback. We are excited that the reviewer found the topic of robustness for UAPs to be important and our methods/experimental results convincing. For our experiments, unless otherwise specified, we will use ILSVRC-2012 with Inception-v3 and our hardest transformation set R(10), T(2,2), Sh(2), Sc(2), B(2, 0.001) and report results with $\gamma = 0.6$. Below we address the comments/questions made by the reviewer.
>
> >**Q1: The contribution of the work is limited/The proposed algorithm has limited novelty.**
>
> **R1:** We have included a response to this in R1 of our general response.
>
> >**Q2: The authors should evaluate the robust UAPs on hold-out transformations, which were not observed during training.**
>
> **R2:** We measure the effectiveness of our robust UAPs against hold-out transformations. In this experiment, we learn a UAP which is robust to $R(5)$ and obtain a robust ASR of $96.2$. We then measure its effectiveness against $Sc(5)$ and get a robust ASR of $85.4$, in contrast, a robust UAP trained directly to be robust to $Sc(5)$ obtains robust ASR of $98.1$. Next, we measure the robustness of UAP trained against $R(5)$ when subjected to transformations from $B(5, 0.01)$. Here we get a robust ASR of $97.3$, whereas a robust UAP trained to be robust to $B(5, 0.01)$ obtains a robust ASR of $99.2$. Finally, we test the robust UAP on $R(5), Sc(5), B(5, 0.01)$ and get a robust ASR of $83.1$. Our previous results show that a UAP trained to be robust against these parameters directly can obtain a robust ASR of $96.1$. In each case, our UAP maintains  robustness on hold-out transformations but has lower performance compared to robust UAPs trained directly to be robust to those transformations. These results are added in Appendix N.
>
> >**Q3: The authors mainly evaluated the robust UAP in the context of untargeted attack success rate. It would be further interesting to see a performance evaluation of the targeted attack success rate.**
>
> **R3:** We can simply turn our algorithm from untargeted to targeted by replacing the loss function. We would like to have target class, A, be classified as target class, B. Instead of maximizing the expected value of the cross entropy loss we can instead formulate the loss based on maximizing B while minimizing A similar to [F]. We randomly select a couple of target classes and perform this attack. Our results are in Appendix O and below. We are measuring our robust ASR of turning class A into class B and observe similar results with RobustUAP being the most robust followed by SGD. It is also interesting to note that different random combinations lead to more or less success, i.e. it is easier to turn a dog into another dog than perfume into a padlock.
>
>
> | Target Class              | Standard UAP | SGD   | Standard UAP_RP | RobustUAP |
> |---------------------------|--------------|-------|-----------------|-----------|
> | Toy Poodle -> Maltese Dog | 42.4%        | 99.1% | 85.6%           | **99.8%**     |
> | Perfume -> Padlock        | 0.0%         | 63.8% | 5.1%            | **76.4%**     |
>
>
> >**Q4: Data efficiency: How do robust UAPs perform when only limited data is accessible during the training process? Please also note that in the literature data-free methods exist.**
>
> **R4:** Here, in order to measure the data efficiency of robust UAPs, we vary the size of the training dataset. The results can be seen in Appendix P. Here we see that RobustUAP is able to obtain relatively robust results with only 500 data points and we get better performance up to around 4000 data points but the results seem to stagnate after that. We are aware of the data-free methods for UAP generation in literature [A], but these methods do not account for robustness. We observe that these methods can obtain similar levels of universal adversarial success rate as UAPs, but similar to standard UAP, they are not robust. We ran [A] on ILSVRC 2012 with the same settings as mentioned in the beginning of this response, they achieve 10.3 robust ASR which is similar to standard UAP.

---

### Official Review · Reviewer_CA3y · 2022-10-24

**Confidence:** 4
**Correctness:** 3
**Technical Novelty And Significance:** 3
**Empirical Novelty And Significance:** 3
**Recommendation:** 5

**Clarity, Quality, Novelty And Reproducibility:**

The paper is well written and the methods are novel. It would be nice to compare with more extensive baselines (see above).

**Strength And Weaknesses:**

Strengths: the method outperforms baselines on adversarial success rate, especially under large sets of transformations.

Weaknesses: the SGD baseline is underspecified. It would be interesting to see the extent to which the presented algorithm outperforms SGD while contextualizing compute used. Some questions:
* How much compute is used to perform RobustUAP compared to SGD?
* How much compute is used to obtain robustness estimates?
* What happens when this same amount of compute is used to run SGD?
* What happens as you vary the number of SGD iterations? Does SGD improve its ASR or does it just asymptote?

**Summary Of The Paper:**

The authors consider the problem of making universal adversarial perturbations (UAPs - those that can be applied to any input and trigger classification for a certain class) under a set of transformations. The authors present a new algorithm. It involves applying SGD over  examples (in the outer loop) and randomly chosen transformations (in the inner loop). The caveat is that for each batch the algorithm estimates robustness (i.e. adversarial success rate under the transformation set) in the optimization loop, allowing the algorithm to move on to the next batch only after the resulting UAPs are estimated to be robust to a certain level. It turns out that in practice this algorithm has higher adversarial success rate than just performing SGD alone, especially on complex sets of transformations.

**Summary Of The Review:**

A well written paper that tackles an important problem with new ideas. The only lacking aspect is extensive baselines. My score will increase if the authors run more extensive baselines.

---

> ### Author Response · Authors · 2022-11-14
> **Response to Reviewer CA3y**
>
> We thank the reviewer for their constructive feedback. We are motivated that the reviewer sees the strength in our approach and the importance of the problem we are trying to solve. Below, we provide additional experiments extensively exploring the SGD baseline in comparison to our RobustUAP algorithm. For our experiments, unless otherwise specified, we will use ILSVRC-2012 with Inception-v3 and our hardest transformation set R(10), T(2,2), Sh(2), Sc(2), B(2, 0.001) and $\gamma=0.6$.
>
> >**Q1: How much compute is used to perform RobustUAP compared to SGD?**
>
> **R1:** RobustUAP uses more compute than SGD. To measure compute, we ran all methods on a desktop PC with a GeForce RTX(TM) 3090 GPU (24 Gb memory) and a 16-core Intel(R) Core(TM) i9-9900KS CPU @ 4.00GHz, on average, RobustUAP takes about 4x as much time to compute (roughly 118 mins compared to 32 mins). Results for other algorithms can be seen in Appendix L.
>
> >**Q2: How much compute is used to obtain robustness estimates?**
>
> **R2:** Robustness estimates depend on a wide variety of factors namely test set size and n. When using n = 738 as presented in the paper, robustness estimate takes on average 19 minutes for the 1000 image test set. While training RobustUAP, EstimateRobustness gets called per batch which takes about 36 seconds for a batch of size 32.
>
> >**Q3: What happens when this same amount of compute is used to run SGD?**
>
> **R3:** We reperform the ILSVRC 2012 experiments presented in Table 1 of the paper. We first compute UAPs for each transformation using SGD. We record the time taken by SGD. We then compute RobustUAP but cut the algorithm off when it hits the time taken by SGD. We present a table of results showing the SGD results, original RobustUAP results, restricted RobustUAP results in Appendix M as well as below. We note that RobustUAP still outperforms SGD (upto 9% more robustness on our most challenging transformation set) at the same compute time, however, it does not perform to the same level as when it is given more time to run.
>
> | Transformation Set                        | SGD   | RobustUAP | Restricted RobustUAP |
> |-------------------------------------------|-------|-----------|----------------------|
> | R(20)                                     | 69.9% | **93.2%**     | 72.9%                |
> | T(2,2)                                    | 96.1% | **97.1%**     | 96.9%                |
> | Sc(5), R(5), B(5, 0.01)                   | 85.4% | **96.1%**     | 86.3%                |
> | R(10), T(2, 2), Sh(2), Sc(2), B(2, 0.001) | 63.1% | **86.4%**     | 72.0%                |
>
> >**Q4: What happens as you vary the number of SGD iterations? Does SGD improve its ASR or does it just asymptote?**
>
>  **R4:** We provide a graph of SGD performance on transformations on ILSVRC 2012 over a number of iterations. In Appendix M, we have provided a graph of the results. We record performance every 50 iterations, we note that SGD is able to quickly improve to a good result, but then stagnates and does not reach the level of performance that RobustUAP does, even after running for the same amount of time (1250 iterations of SGD is approximately the time it takes for RobustUAP to finish).
>
> We hope that these experiments help to establish a better baseline for comparison. We have additionally provided more experiments in the appendix as suggested by other reviewers and hope that these will provide further empirical evidence for the strength of our method.

---

### Official Review · Reviewer_BeFR · 2022-10-25

**Confidence:** 4
**Correctness:** 3
**Technical Novelty And Significance:** 2
**Empirical Novelty And Significance:** 2
**Recommendation:** 5

**Clarity, Quality, Novelty And Reproducibility:**

It is well written and easy to follow.
It is limited in its novelty and experimental evaluation, as mentioned above.

**Strength And Weaknesses:**

positives:
+ it is important to improve the robustness of universal perturbation, especially when applying them in physical attack.


negatives:
- the proposed method is straightforward since it just combines the idea of physical attack and digital universal perturbation attack.
- But experiments conducted in this paper almost focus on the digital domain, instead of real-world physical transformation. Due to that, it is hard to measure the effectiveness of the proposed method.

**Summary Of The Paper:**

This paper proposes a robust universal adversarial perturbation which is robust to many transformations such as rotation, pixel intensity. It proposes a robustness estimation approach which can be leveraged to conduct adversarial attack.

**Summary Of The Review:**

The problem discussed in this paper is important when applying universal perturbation attack to real-world physical attack. But my concerns are the limited novelty and experimental evaluation, thus it needs to be significantly improved before being accepted.

---

> ### Author Response · Authors · 2022-11-14
> **Response to Reviewer BeFR**
>
> We thank the reviewer for their constructive feedback. We are motivated that the reviewer agrees on the importance of robustness for universal perturbations, especially for real-world, physical attacks. Below, we address the comments of the reviewer.
>
> > **Q1: The proposed method is straightforward since it just combines the idea of physical attack and digital universal perturbation attack.**
>
> **R1:** We have included a response to this in R1 of our general response.
>
>  > **Q2: Experiments conducted in this paper almost focus on the digital domain, instead of real-world physical transformation. Due to that, it is hard to measure the effectiveness of the proposed method.**
>
> **R2:** As mentioned in the introduction, many real-world attacks require generating UAPs robust against a set of transformations that an adversary thinks can modify the UAP during transmission. These transformations are specific to the considered attack scenario and require significant modeling and expertise. Instead of creating specific attacks, we focus on designing general, robust algorithms that can be combined with domain-specific modeling of transformations to create fundamentally stronger real-world attacks than possible with existing algorithms.
>
> We consider an additional real-world transformation which adds fog to an image based on [A] in Appendix K in the updated paper. The graph in Appendix K shows similar results to what we have observed for other transformation sets. We will be happy to provide robustness results against any other real-world transformations that the reviewer may have in mind.
>
> We have also provided a number of additional experiments as requested by other reviewers and hope that these help to bulk up the experimental evaluation. We are happy to run any other experiments.
>
> [A] 3D Common Corruptions and Data Augmentation; CVPR 2022

---

### Author Response · Authors · 2022-11-14
**General Response**

Dear Area Chair and Reviewers,

We thank all the reviewers for their constructive comments and have updated our paper accordingly (changes in blue). In our general response, we respond to a common important question: novelty. The reviewers asked for more extensive experiments. We have performed all of them. We provide a list of the new experiments and where to find their results in the Appendix here.

> **Q1: The proposed method is straightforward since it just combines the idea of physical attack and digital universal perturbation attack (BeFR). The contribution of the work is limited/The proposed algorithm has limited novelty (jAxg).**

**R1:**  Our baselines, SGD and Standard UAP with robust perturbations, combine both the ideas of physical attacks and digital UAPs. As we show with our experimental results, these baselines achieve suboptimal results and are fundamentally limited in the amount of robustness they can offer (see our study on the robustness of the SGD baseline after increasing the number of iterations in Appendix M added in response **R4** to reviewer CA3y). This is because the approximation of the expectation in eq. (5) computed by these methods is too imprecise. In contrast, in our method, we rely on a novel sampling strategy that can provide probabilistic bounds on the closeness between the computed and theoretical expectations in eq. (5). This enables better robustness of the generated UAPs as shown in our experimental results.

These results confirm that not all combinations of EoT, batch-training, PGD, and UAP obtain the same level of robustness. While existing works such as [A,B,C] show SGD based methods for UAP creation they do not address robustness of UAPs or provide probabilistic guarantees for robustness. As confirmed by our experimental results, simply using SGD is not enough to get both robustness to transformation and universality.

> **Additional Experiments Provided**
 - Appendix K - Exploring more real-world corruptions like Fog
 - Appendix L - Algorithmic Runtimes
 - Appendix M - Effect of Compute Time on Robustness
 - Appendix N - Robustness on Hold-Out Transformations
 - Appendix O - Targeted Attacks
 - Appendix P - Data Efficiency
 - Appendix Q - Transferability
 - Appendix R - Transformer-Based Models
 - Appendix S - Robust UAPs on Robust Networks
 - Appendix T - Ablation on Optimization Strategy

We believe that the extra experimental evidence added above should be enough to convince the reviewers that our method ensures extra robustness not possible with existing methods. This is due to explicit formulation of robustness within a perturbation region in our optimization objective. If any other experiments might interest you please let us know, we would also be happy to make any other small or big changes you may think of. Thank you all for your time!

[A] Generalizable Data-free Objective for Crafting Universal Adversarial Perturbations; T-PAMI 2018

[B] Understanding Adversarial Examples from the Mutual Influence of Images and Perturbations; CVPR 2020

[C] Universal Adversarial Training; AAAI 2020

---

### Author Response · Authors · 2022-11-17
**Final Questions/Comments**

Dear Reviewers,

As the discussion period is coming to a close, we would like to take the opportunity to once again thank you for your time. We hope the comments we have made so far have addressed all prior concerns/questions, but please feel free to let us know if there is anything we can elaborate on. If there are any other experiments or questions you have please let us know and we will do our best to run/address anything before the deadline.

---

### Author Response · Authors · 2022-12-13
**Further Discussion**

Dear Area Chair,

Would you mind to encouraging some discussion for our paper (we have only received one comment after the rebuttal)? We believe we have responded to all of the reviewers existing comments, but we would love to respond to any of your further questions after your internal discussion.

---

### Decision · Program_Chairs · 2023-01-20

**Decision:**

Reject

**Justification For Why Not Higher Score:**

The major concerns are that (1) as mentioned by Reviewer CA3y and responded by authors, Restricted RobustUAP is slightly better than SGD. However, this phenomenon is not clearly described in the paper and the results shown in the tables will give a false sense that RobustUAP is significantly better than SGD under the same condition. (2) Some issues, including data-free methods for UAP generation and targeted attacks, raised by the reviewers, need to further explore instead of providing few (but selective?) experimental results. (3) As suggested by Reviewer pw3U, the paper is tougher than it needs to be to read because of the wall of theory in front of a very simple method.

**Justification For Why Not Lower Score:**

N/A

**Metareview: Summary, Strengths And Weaknesses:**

In this paper, the authors studied the robust universal adversarial perturbation (UAP) problem.
The proposed concept is simple but the novelty, as claimed by the authors, is that they rely on a novel sampling strategy that can provide probabilistic bounds on the closeness between the computed and theoretical expectations in eq. (5). This enables better robustness of the generated UAPs as shown in the experimental results.
The authors also confirm that not all combinations of EoT, batch-training, PGD, and UAP obtain the same level of robustness.

The major concerns are that (1) as mentioned by Reviewer CA3y and responded by authors, Restricted RobustUAP is slightly better than SGD. However, this phenomenon is not clearly described in the paper and the results shown in the tables will give a false sense that RobustUAP is significantly better than SGD under the same condition. (2) Some issues, including data-free methods for UAP generation and targeted attacks, raised by the reviewers, need to further explore instead of providing few (but selective?) experimental results. (3) As suggested by Reviewer pw3U, the paper is tougher than it needs to be to read because of the wall of theory in front of a very simple method.
Finally, how to select the parameter, \gamma (0.5/0.6(Table 1)/0.7(Table 4)/0.8(Table 2)), is not clear.

Based on the above reasons, this paper indeed has some merits but needs more work to make it complete.
It is suggested to be rejected at its current status.